# Effects of Duodenal 5-Hydroxytryptophan Perfusion on Melatonin Synthesis in GI Tract of Sheep

**DOI:** 10.3390/molecules26175275

**Published:** 2021-08-31

**Authors:** Jun Pan, Fengming Li, Caidie Wang, Xiaobin Li, Shiqi Zhang, Wenjie Zhang, Guodong Zhao, Chen Ma, Guoshi Liu, Kailun Yang

**Affiliations:** 1College of Animal Science, Xinjiang Agricultural University, Urumqi 830052, China; 320180040@xjau.edu.cn (J.P.); lifming@xjau.edu.cn (F.L.); caidie5338352@163.com (C.W.); lxb262819@163.com (X.L.); zhangas_q@163.com (S.Z.); XNDzhangwenjie@163.com (W.Z.); Guodong_Zhao@163.com (G.Z.); machen009@126.com (C.M.); 2College of Animal Science and Technology, China Agricultural University, Beijing 100083, China; gshliu@cau.edu.cn

**Keywords:** sheep, intestinal mucosa, 5-hydroxytryptophan, melatonin, gastrointestinal track

## Abstract

The purpose of this study is to investigate the potential effects of 5-hydroxytryptophan (5-HTP) duodenal perfusion on melatonin (MT) synthesis in the gastrointestinal (GI) tract of sheep. 5-hydroxytryptophan is a precursor in the melatonin synthetic pathway. The results showed that this method significantly increased melatonin production in the mucosa of all segments in GI tract including duodenum, jejunum, ileum, cecum and colon. The highest melatonin level was identified in the colon and this indicates that the microbiota located in the colon may also participate in the melatonin production. In addition, portion of the melatonin generated by the GI tract can pass the liver metabolism and enters the circulation via portal vein. The current study provides further evidence to support that GI tract is the major site for melatonin synthesis and the GI melatonin also contributes to the circulatory melatonin level since plasma melatonin concentrations in 5-HTP treated groups were significantly higher than those in the control group. In conclusion, the results show that 10–50 mg of 5-HTP flowing into the duodenum within 6 h effectively improve the production of melatonin in the GI tract and melatonin concentration in sheep blood circulation during the day.

## 1. Introduction

Melatonin (MT) has multiple biological functions in mammals, including sleep promotion, immunoregulation, and antibacterial, anti-inflammatory, as well as antioxidative activities [1,2,3,4]. Since the first isolation of MT from bovine pineal gland [5], MT was believed to be exclusively produced in the pineal gland until MT was detected in the retina, Harderian gland, cerebellum [6] and in the intestinal chromaffin cells [7]. These observations have expanded the melatonin synthetic sites to extrapineal tissues and organs. Currently, it has been reported that melatonin is synthesized by mitochondria and many functions of melatonin is also mediated by the mitochondria [8]. As a result, virtually, every cell with mitochondria has the capacity to synthesize melatonin. For example, the total amounts of MT produced in the gastrointestinal tract (GI Tract) were estimated to be approximately 400 times more than that produced in the pineal gland [9]. As to its synthesis, tryptophan (Trp) is its original precursor of melatonin in mammals [10]. First, Trp is decarboxylated to 5-hydroxytryptophan (5-HTP) by the tryptophan hydroxylase (TPH) [11], and then 5-HTP is transformed into serotonin (5-HT) by aryl amino acid decarboxylase (AADC), and 5-HT is converted to N-acetyl-5-hydroxytryptamine by a rate-limiting enzymes, arylalkylamine-*N*-acetyltransferase (SNAT, previously AANAT), finally, the N-acetyl-5-hydroxytryptamine is converted to melatonin by N-acetyl-5-hydroxytryptamine methyltransferase (ASMT, previously HIOMT) [12]. Melatonin synthetic pathway in mammals is illustrated in the Figure 1.

MT plays important roles in the improvement of the yields and quality of animal products. For example, it improves the meat quality and quantity of cashmere goats [13,14], reduces somatic cell count and mastitis in dairy cattle to improve the quality of the milk [15,16]. Currently, in husbandry industrial, the methods used to regulation of MT levels in animals are primarily to alter the light/dark cycle of the environments [17,18], or by implantation [19,20], injection [21], and oral administration [22] of MT. As to the exogenous melatonin administration, several factors should be considered including its bioavailability, acting duration, economic benefit, and delivery methods [22]. Additionally, exogenous MT supplementation seems to reduce endogenous MT synthesis and to down-regulate expression of the MT receptors [23,24,25]. Thus, researchers have tried to improve melatonin production by manipulating the availability of MT precursor, that is, to give Trp to animals. The results are variable. Some studies indicated that only 1–2% of Trp ingested were converted into 5-HT depending on the species [26]. Several studies have shown that Trp supply increases blood MT levels in the tested animals [27,28,29,30]. However, this increase only occurs nocturnally, or at certain time points [31,32], while others reported the negative results with no increased melatonin production [33,34]. Specifically to sheep, a study showed that 5-HTP supply increased the MT synthesis, but Trp supply failed to achieve this goal [35]. Due to the uncertainty of Trp supply on melatonin production in sheep, we select the 5-HTP as the precursor to feed sheep and attempt to identify whether this treatment can improve the melatonin production in sheep. Indeed, we have found that feeding 5-HTP increases MT content of blood plasma from the jugular vein in sheep [36,37] which mainly reflects the pineal originated melatonin [9,38]. Few studies have investigated the effects of 5-HTP supply on the GI tract melatonin synthesis [39], especially, the different segments of GI tract. GI tract is considered as one of the largest extrapineal melatonin generating site and melatonin plays the important physiological functions on it [10]. Thus, in the current study, we focus whether 5-HTP supplementation also increases the melatonin synthesis in the different sections of GI tract. As we know that the melatonin in portal vein blood plasma is the indicator of GI generated melatonin and the melatonin in jugular vein mainly represents the pineal generated melatonin [9,38]. For comparative purpose, both the portal vein and the jugular vein blood were collected. The results from the current study will answer the following questions: (1) Whether duodenal 5-HTP supplementation will increase the GI melatonin synthesis? (2) If it is so, does the GI synthesized melatonin contribute to the circulatory melatonin?

## 2. Results

### 2.1. Effects of Different Doses of 5-HTP Duodenal Perfusion on 5-HTP Content in Sheep Intestinal Mucosa

The levels of 5-HTP in general intestinal tract mucosa were significantly increased in both 10 and 50 mg duodenal 5-HTP perfused groups compared to the control group (*p* < 0.01) (Figure 2A). 5-HTP contents in the jejunum, ileum, cecum, and colon mucosa were increased by 35.56% (*p* < 0.01), 30.19% (*p* < 0.01), 2.85% (*p* > 0.05), and 22.40% (*p* < 0.05), respectively, in the 10 mg 5-HTP perfused group compared to controls (Figure 2B). In the 50 mg group, the 5-HTP levels in duodenum, jejunum, ileum, cecum, and colon mucosa increased by 44.17 (*p* < 0.01), 28.00% (*p* < 0.05), 30.26% (*p* < 0.01), 30.53% (*p* < 0.01), and 28.14% (*p* < 0.05), respectively. The highest concentration of 5-HTP in 50 mg group was observed in the duodenum mucosa (Figure 2B).

### 2.2. Effects of Different Doses of 5-HTP Duodenal Perfusion on the Blood 5-HTP Levels Collected from Different Blood Vessels

The 5-HTP concentrations in circulation blood plasma were significantly increased with 10 and 50 mg 5-HTP duodenal infusion (*p* < 0.01) (Figure 3A). It seemed that the rapid increased 5-HTP concentrations in circulation blood also quickly reached its steady status within one hour after perfusion (Figure 3B).

In addition to the circulation blood, the 5-HTP concentrations in blood of portal vein, carotid artery, jugular vein, and posterior vena cava were also significantly increased compared to their respective controls (*p* < 0.01) (Figure 4A–D). While 5-HTP concentrations among the perfused groups did not exhibit significant differences during the process of 5-HTP duodenal perfusion (*p* > 0.05) (Figure 4A–D).

As to the timely dynamic changes of plasma 5-HTP concentration from portal vein, the 5-HTP level rapidly increased to 257.5 ng/mL after 1 h of perfusion, and then stabilized at the range from 212.9 to 242.5 ng/mL during the entire experimental period and its level at 4 h was significantly higher than that of the control (*p* < 0.05). However, 2 h after completion of perfusion, the concentrations of 5-HTP were declined in both of two 5-HTP perfusion groups. In addition, no significant differences were observed in the two 5-HTP perfusion groups. The very similar pattern of the plasma 5-HTP was observed in the blood from carotid artery, jugular vein, and posterior vena cava, respectively (Figure 5B–D).

### 2.3. Effects of Different Doses of 5-HTP Perfusion on MT Content in Sheep Intestinal Mucosa

The contents of MT in entire intestinal tract mucosa were significantly increased by 23.16% (*p* < 0.05) and 49.70% (*p* < 0.01) in 10 and 50 mg duodenal 5-HTP perfused groups, respectively compared to the control (Figure 6A). MT contents in different GI tract segments also significantly increased including jejunum (*p* < 0.05), ileum (*p* < 0.01), cecum (*p* < 0.01), and colon (*p* < 0.01) mucosa in 50 mg 5-HTP duodenal perfused group compared to the control group (Figure 6B), while MT content levels in the ileum (*p* < 0.01) and cecum (*p* < 0.05) mucosa in 10 mg 5-HTP perfused groups were also significantly increased compared to their control group. The highest content of MT was found in the colon mucosa of 50 mg 5-HTP perfused group (around 7.8 pg/mg tissue).

### 2.4. Effects of Different Doses of 5-HTP Duodenal Perfusion on Blood MT Levels from Different Blood Vessels

The circulating blood plasma MT concentrations were significantly increased in 10 and 50 mg 5-HTP duodenal perfused groups, respectively, compared to the control group (*p* < 0.01), and MT concentration in 50 mg group was also higher than that in 10 mg group (*p* < 0.05). The highest melatonin concentration in the circulating blood was achieved at 4 h after perfusion and the much higher melatonin concentrations were preserved until 2 h after the ending of perfusion (Figure 7B).

The blood plasma MT concentrations in the portal vein, carotid artery, jugular vein and posterior vena cava, were also significantly higher in the 10 and 50 mg treated groups than those in their respective control group (*p* < 0.01) (Figure 8A–D). In addition, plasma MT concentrations in the 50 mg group were also higher than those of the 10 mg group in the portal vein, carotid artery, jugular vein and posterior vena cava, respectively (*p* > 0.05).

The timely dynamic changes of blood melatonin concentrations in the different blood vessels were also monitored. For example, in the portal vein, the highest blood melatonin concentration (63.5 pg/mL) was achieved at the 4 h after 5-HTP perfusion which were higher than that of control group (*p* < 0.05), and these levels fluctuated in the range of 48.9 to 63.5 pg/mL during the entire perfusion period until to the 2 h after perfusion terminated, while these levels of control were 40.9 to 46.9 pg/mL (Figure 9A). The blood melatonin concentrations in the carotid artery, jugular vein and posterior vena cava showed the similar pattern as in the portal vein (Figure 9B–D).

## 3. Discussion

Whether the Trp supplementation can increase melatonin production in sheep is debatable [33]. Thus, in the current study, we avoided using Trp but selected another precursor of melatonin synthesis, 5-HTP, to test its effect on melatonin productions in sheep (Kazakh sheep). As we know that sheep are ruminants, if 5-HTP is orally given to sheep, it needs to pass the rumen in which digestive enzymes and microbiota may consume some of the 5-HTP. This will impact the bioavailability of 5-HTP. In this consideration, we have selected the duodenal perfusion to deliver the 5-HTP. The results showed that duodenal perfusion of 5-HTP significantly increased the blood and GI tissue 5-HTP levels. This observation is consistent with previous reports. Studies have shown when sheep were orally given 111–222 mg/kg LBW (equivalent 5-HTP 50–100 mg/kg LBW) rumen-protected 5-HTP, it increased their jejunum, ileum, cecum, and colon mucosa and plasma 5-HTP concentrations; however, the dose they used was much higher than those used in this study [36,39,40]. It has been reported that high dose of 5-HTP intravenous administration (over 300 mg/d) leads to animal developing poisoning symptoms similar to 5-HT-induced toxicity [41]. In the pilot study, we have also found that if the 5-HTP duodenal perfusion dose in excessive of 50 mg/d (equivalent 1 mg/kg LBW) the sheep drastically reduced food intake or even refused to eat, and then, the sheep developed irritable and restless symptoms. While if the dose of 5-HTP was under 50 mg/d, no adverse effects were observed. Based on the results mentioned above, in the current study, 5-HTP doses selected were 10 and 50 mg/d which is 50–100 times lower than that used by Zhao et al. [36,39]. This difference may be due to the bioavailability of 5-HTP by the different delivery methods. 5-HTP orally administration has to pass the rumen and majority of it may be metabolized by the digestive enzymes or microbiota inhabited in the rumen.

5-HTP is a direct precursor of 5-HT and it can be transformed into 5-HT by aryl amino acid decarboxylase, without biochemical feedback or rate-limiting steps. In GI tract, this process mainly occurs in the intestinal chromaffin cells of the intestinal mucosa [40,42,43]. The increased concentration of 5-HT in the small intestine after 5-HTP treatment has also been reported in different species including rats [42] and humans with Parkinson’s disease [44] or health subjects [45]. Our focus is whether this increased 5-HTP levels in local GI tissue and blood will be converted to melatonin since there is few investigations to deal this important question. Our results, for the first time, showed that the increased 5-HTP not only elevated the melatonin content in the GI tissue mucosa but also increase the circulatory melatonin levels such as in the blood from carotid artery. This is a long-lasting argument whether the GI track synthesized melatonin is attributed to the circulatory melatonin. It has been hypothesized that the locally synthesized melatonin would not release into the circulation but be consumed by the local tissues as the antioxidant [46]. Our results clearly showed that the GI track locally synthesized melatonin contribute to the circulatory melatonin.

By collecting blood from different blood vessels, we try to identify the sources of melatonin in the blood. We observed in all animals, their melatonin levels in jugular vein are higher than that in the carotid artery. This indicated that brain tissue releases melatonin into blood. Several studies have also investigated the potential relationship between the melatonin levels of jugular vein and carotid artery. The results are not consistent. Bubenik et al. reported that 1–2 h after feeding, jugular vein melatonin only slightly increased with no significant difference with the melatonin in the carotid artery [38]. This is similar to the observation of Huether et al. with Trp orally administration [9]. In contrast, nocturnal concentration of MT was reduced after intramuscular injection of 5-HTP in quails [47]. This discrepancy may be species specific. In sheep, with intraperitoneal injection of 5-HTP the diurnal MT level in jugular vein was increased but no effect was observed on nocturnal MT level [35,48]. As to the melatonin levels in the portal vein, the results are consistent based on the previous observations. In pigs, the highest melatonin level was observed in the portal vein among carotid artery and posterior vena cava after feeding [38]. In chicken and rats, after oral administration of 300 mg/kg Trp, the portal vein melatonin was significantly higher than that in jugular vein [9]. Similarly, the concentration of MT in human portal vein was also higher than that in peripheral veins [49]. Our results are in consistent with the above-mentioned results. It is well documented that major portion of melatonin is metabolized in the liver by the CYP 1A2 to form the 6-hydroxymelatonin [50]. For example, studies reported that 92–97% of MT in male Sprague–Dawley rats was cleared up by the liver via a single passage [51]. This strongly suggest that much more melatonin has been produced in the GI tract before it reaches to the portal vein. Originally, intestinal chromaffin cells are believed to be the main site of intestinal MT synthesis [7]. The AA-NAT and HIOMT identified in rat intestinal chromaffin further support this observation [52].

However, following the discovery of mitochondria being the main site of melatonin synthesis [53], virtually all the cells in the GI tract have the capacity to synthesize melatonin. Our observations that mucosa of all the segments in GI tract including duodenum, jejunum, ileum, cecum and colon have increased melatonin content after 5-HTP perfusion support this idea. Very interestingly, the highest melatonin level was identified in the colon. This is not surprising due to the discovery that the microorganisms including the bacteria have the ability to synthesize melatonin [54], since the microbiota located in the colon also participate in the melatonin synthesis. The melatonin produced in GI tract is primarily functional locally to protect the gut from the oxidative stress and inflammation [55,56], but also can release into circulation for other systemic functions. The limitation of this study is that we have not explored whether the GI generated melatonin contribute to the normal melatonin circadian rhythm. This requires collecting blood samples during a 24 h interval. In the current study, the blood samples were only collected during an 8 h interval. However, to test the contribution of GI generated melatonin to melatonin circadian rhythm will be our future project. In conclusion, the results show that 5-HTP duodenal perfusion avoids the metabolism of this substance by the digestive enzymes and microbiota in the rumen and increases the bioavailability of 5-HTP. Therefore, duodenal supplementation with 10–50 mg 5-HTP within 6 h can effectively improve the production of melatonin in sheep, particularly in the GI tract of the sheep.

## 4. Material and Methods

### 4.1. Materials

5-HTP (purity, >98%) was purchased from Xi’an lvruquan Biotechnology Co., Ltd. (Xi’an, China).

#### 4.1.1. Animals

Local Kazakh sheep (*Ovis*
*aries*) from Xinjiang, China, were selected and all animals were housed in the animal farmhouse. The experiments were conducted in Changji Huikang Animal Husbandry Co., Ltd., Xinjiang, China, from April to June in 2020. During the experiment period, the local sunrise times were 06:23–06:41, sunset times were 21:33–21:56, the external environment temperature was 22–33 °C, and the temperature in the animal house was 25–30 °C. The animal study has been approved by the Animal Welfare and Ethics Committee of Xinjiang Agricultural University, and the approved protocol number was 2020022.

#### 4.1.2. Experimental Design

Kazakh sheep (*n* = 18), aged 15–18 months and weighing (49.80 ± 1.96 kg) were used in the experiment. Six months after the installation of duodenal fistula, the 18 sheep were divided into three groups with even numbers using a single factor randomized block design based on the sheep body weight, as follows: Control group: duodenal perfusion with 400 mL normal saline; 5-HTP treated group I and II: duodenal perfusion with 400 mL normal saline containing 10 mg and 50 mg 5-HTP, respectively.

#### 4.1.3. Animal Feeding and Management

The sheep were kept in a single/pen (i.e., 18 pens in total) with free access to drinking water. Sheep diet composition and nutrition levels (presented in Table 1) were formulated according to National Research Council (NRC) version 2007 (NRC07). Each sheep was fed with concentrate formula (600 g/d), alfalfa hay (500 g/d), and wheat straw (300 g/d). The roughage was crushed to 2–3 cm and mixed in proportion. The diet was fed with roughage first, followed by concentrate formula. For the first 6 days, sheep were fed twice a day, and on the seventh day, they were deprived of diet and water from 6:00 a.m.

### 4.2. Duodenal Perfusion and Sample Collection

#### 4.2.1. Duodenal Perfusion

5-HTP (10 mg = 0.02 mg/kg LBW and 50 mg = 1 mg/kg LBW) were accurately weighed and dissolved in 400 mL normal saline. The solution was perfused via the duodenal fistula at a rate of 1.11 mL/min (22 drops/min) beginning at 7:30 a.m. after first feeding. The perfusion flow rate was checked and corrected every 30 min, to ensure the stable flow. The perfusion process lasted approximately 6 h/d and for consecutively 7 days.

#### 4.2.2. Animal Surgery and Blood Collection

At 6:00 a.m. of day 7, the animals were anesthetized with 0.02 mL/kg xylazine hydrochloride, a 15–20 cm abdominal incision was made at 1 cm from the posterior edge of the thirteen ribs, and the hepatic portal vein, posterior vena cava, carotid artery, and jugular vein were isolated and catheterized for blood collection. The methods were briefly described as following. After abdominal incision, the hepatic portal lymph node was located and a trocar inserted into the hepatic portal vein at a 45° angle anterior to the hepatic portal lymph node. Then, the internal needle was withdrawn and the cannula inserted 1–2 cm forward, with the tip of the cannula 0.5 cm away from the liver. A similar method was used to insert the posterior vena cava cannula, with the tip located below the head of the kidney. An incision was made in the neck skin and the jugular vein and carotid artery intubated in the same way, with the tip of the cannula located 10 cm away from the head. The blood was collected at 7:30 a.m. before duodenal perfusion. Then, it started the regular 6 h of 5-HTP duodenal perfusion. After the completion of the 6 h of perfusion, 5 mL of blood were collected from each of the four blood vessels mentioned above at the time points of 1, 2, 3, 4, 5, 6, 7, and 8 h, respectively. The blood samples were rapidly transferred to heparin sodium anticoagulant tubes and centrifuged at 1350× *g* for 15 min, plasma samples were collected into cryopreservation tubes and stored at −20 °C for future use to detect the MT and 5-HTP concentrations. Blood and mucosa sample collection procedure are illustrated in Figure 10.

#### 4.2.3. Mucosal Sample Collection

Two hours after completion of 6 h of 5-HTP perfusion, the two ends of the junction of abomasum and rectum were ligated with sutures under anesthesia. Then, the sheep intestines were quickly removed, and the duodenum, jejunum, ileum, cecum, colon, and rectum were excised, respectively. The contents of each intestinal segment were gently transferred into a clean beaker. Then, each intestinal segment was cut longitudinally and washed five times with normal saline, and the surface of the mucosa was dried by torching to filter paper. The intestinal mucosa was scraped with a slide, and then transferred into an RNAse-free cryopreservation tube, and immediately storage at liquid nitrogen for future use.

### 4.3. Measurement of 5-HTP and MT

Plasma sample pretreatment: After thawing, plasma samples were fully mixed, centrifuged at 1000× *g* at ambient temperature for 10 min, and then supernatants were collected for measurement.

Intestinal mucosa sample pretreatment: After thawing, mucosa sample were centrifuged at 1000× *g* for 10 min, and cleaned with filter paper to remove the remaining blood and fluid. 1 mL normal saline, containing 20 µL of 0.05 mol/L acetic acid were added to the sample and fully homogenized with a homogenizer and then, sonicated with ultrasound for 20 min. The samples were centrifuged at 1000× *g* at ambient temperature for 10 min, the supernatants were collected. The pH of the supernatant was adjusted to 7.4 using 0.05 mol NaOH, and a 10 µL aliquot were used for protein quantification. One gram protein of each sample was used for plasma and intestinal mucosa MT and 5-HTP determination, which were measured by enzyme immunoassay using an ELISA kit (Beijing Sinouk Institute of Biological Technology, Beijing, China) analyzed with ELISA analyzer (Waldron dr-200 Bs) following the manufacturer’s instructions.

### 4.4. Statistical Analyses

All data were analyzed using General Linear Model Procedure of repeated-measurement ANOVA with SAS statistical software (SAS 9.4, SAS Institute, Cary, NC, USA). Different doses of 5-HTP, different vessels of 5-HTP and MT concentration, and each time points of 5-HTP and MT concentration were analyzed separately with the model of Y = D + V + T + D * V + D * T + D * V * T. Here, D represents dosage, V represents vessels, T represents time. The least square means (LSM) and standard error of the mean (SEM) were obtained from the model, the multiple comparisons with LSM were performed with PDIFF option. All data were expressed as LSM ± SEM. The statistical significance was set up *p* < 0.05, highly significant *p* < 0.01.

## Figures and Tables

**Figure 1 molecules-26-05275-f001:**
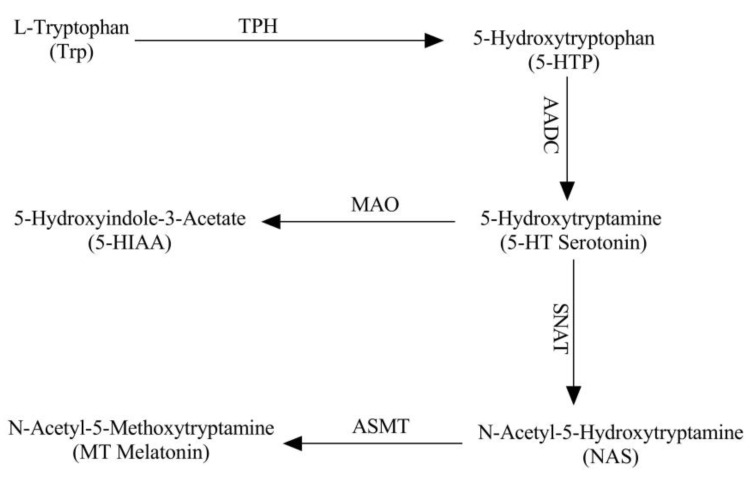
Melatonin synthetic pathway in mammals. TPH: tryptophan hydroxylase, AADC: aryl amino acid decarboxylase, SNAT: serotonin N-acetyltrasferase, ASMT: N-acetyltraptamine methyltrasferase, MAO: monoamine oxidase.

**Figure 2 molecules-26-05275-f002:**
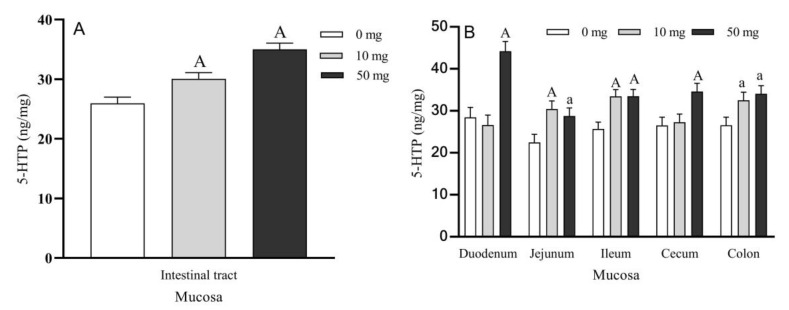
Effects of different doses of 5-HTP duodenal perfusion on 5-HTP content in sheep intestinal mucosa. (**A**) 5-HTP levels in general intestinal tract mucosa. (**B**) 5-HTP levels in the different intestinal segment mucosa. The data were expressed as LSM ± SEM (*n* = 6). Small letters indicate *p* < 0.05, capital letters indicate *p* < 0.01 vs. their respective control groups.

**Figure 3 molecules-26-05275-f003:**
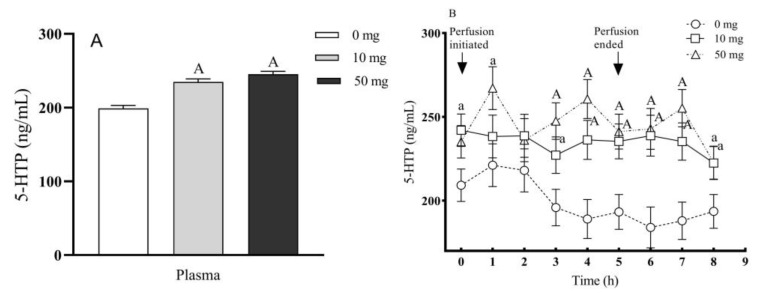
Effects of different doses of 5-HTP duodenal perfusion on circulating blood 5-HTP content. (**A**) 5-HTP levels in the circulating blood plasma. (**B**) Dynamic changes of 5-HTP levels in the circulating blood plasma with time. The data were expressed as LSM ± SEM (*n* = 6). Small letters indicate *p* < 0.05, capital letters indicate *p* < 0.01 vs. their respective control groups.

**Figure 4 molecules-26-05275-f004:**
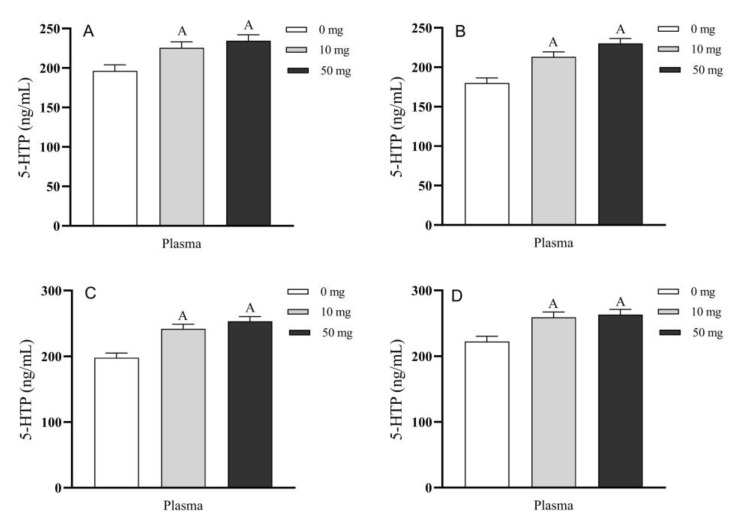
Effects of different doses of 5-HTP duodenal perfusion on blood 5-HTP levels of different blood vessels. (**A**) portal vein, (**B**) carotid artery, (**C**) jugular vein and (**D**) posterior vena cava. The data were expressed as LSM ± SEM (*n* = 6). The capital letters indicate *p* < 0.01 vs. control.

**Figure 5 molecules-26-05275-f005:**
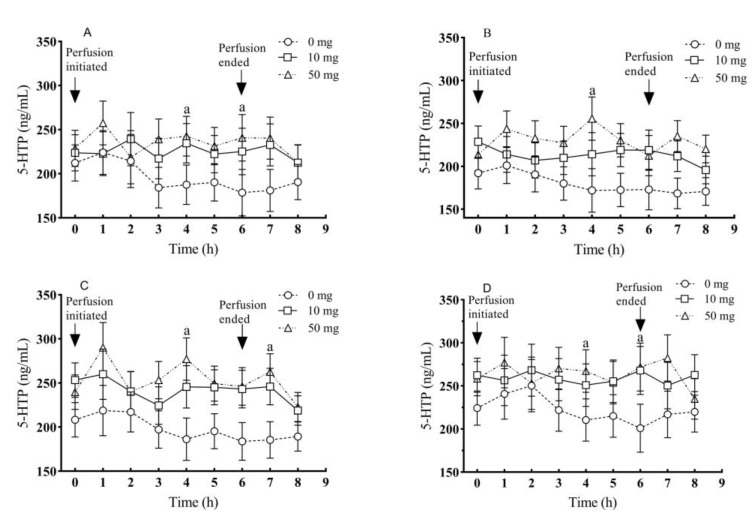
Effects of different levels of duodenal 5-HTP perfusion on the timely dynamic changes of blood 5-HTP in different blood vessels: (**A**) portal vein, (**B**) carotid artery, (**C**) jugular vein and (**D**) posterior vena cava. The data were expressed as LSM ± SEM (*n* = 6). Small letters indicate *p* < 0.05 vs. their respective controls.

**Figure 6 molecules-26-05275-f006:**
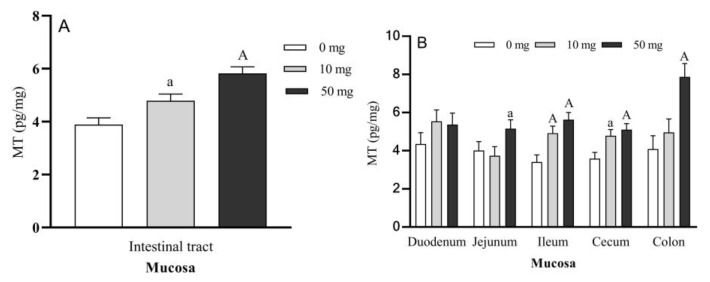
Effects of different doses of 5-HTP duodenal perfusion on MT content in sheep intestinal mucosa. (**A**). MT level in entire intestinal tract mucosa, (**B**). MT levels in mucosa of duodenum, jejunum, ileum cecum and colon, respectively. The data were expressed as LSM ± SEM (*n* = 6). Small letters indicate *p* < 0.05, capital letters indicate *p* < 0.01 vs. their respective control groups.

**Figure 7 molecules-26-05275-f007:**
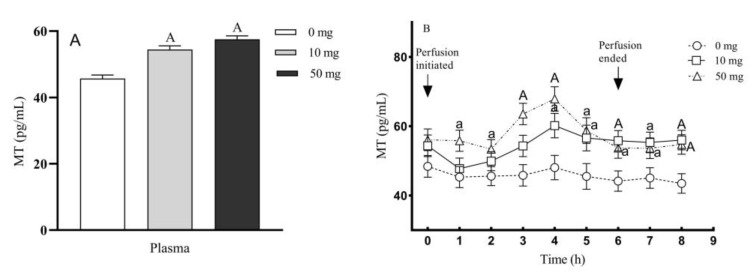
Effects of different doses of 5-HTP duodenal perfusion on circulating blood MT concentrations. (**A**) Circulating blood melatonin concentration. (**B**) The timely dynamic changes of circulating blood MT concentration after 5-HTP duodenal perfusion. The data were expressed as LSM ± SEM (*n* = 6). Small letters indicate *p* < 0.05, capital letters indicate *p* < 0.01 vs. their respective control groups.

**Figure 8 molecules-26-05275-f008:**
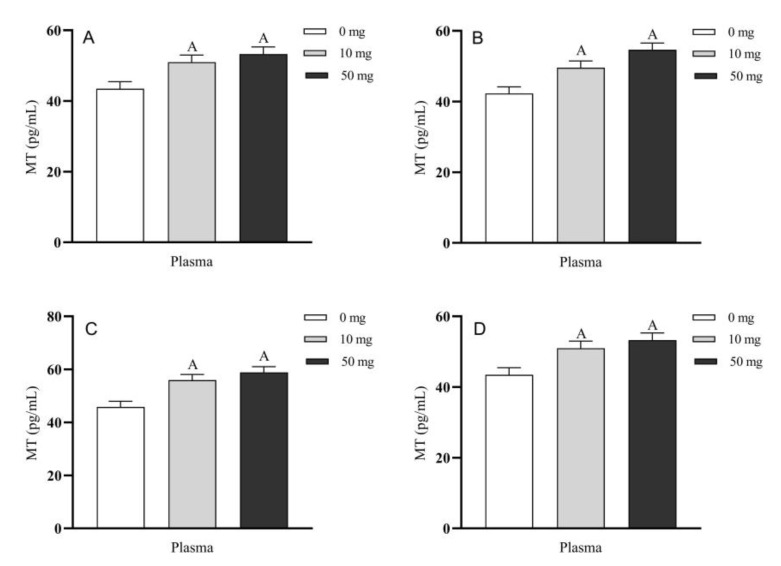
Blood MT concentrations in different blood vessels after 5-HTP duodenal perfusion. (**A**) portal vein, (**B**) carotid artery, (**C**) jugular vein and (**D**) posterior vena cava. The data were expressed as LSM ± SEM (*n* = 6). The capital letters indicate *p* < 0.01 vs. control.

**Figure 9 molecules-26-05275-f009:**
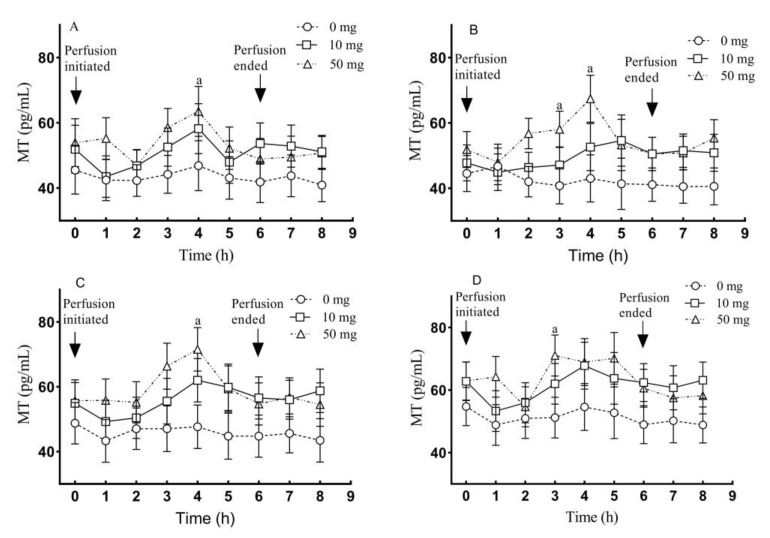
The timely dynamic changes of melatonin concentration in different blood vessels after 5-HTP duodenal perfusion. (**A**) Portal vein, (**B**) carotid artery, (**C**) jugular vein and (**D**) posterior vena cava. The data were expressed as LSM ± SEM (*n* = 6). Small letters indicate *p* < 0.05 vs. their respective control groups.

**Figure 10 molecules-26-05275-f010:**
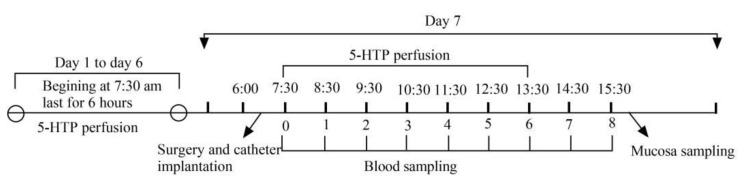
Blood and mucosa sample collection procedure.

**Table 1 molecules-26-05275-t001:** Diet composition and nutrient levels (%).

Ingredient	Ratio	Items	Nutrient Levels
Maize	25.00	Dry matter DM	92.01
Wheat bran	6.00	Crude protein CP	16.39
Soybean meal	10.00	Ether extract EE	9.91
Cottonseed meal	6.50	Neutral detergent fiber NDF	52.89
Alfalfa hay	33.33	Acid detergent fiber ADF	36.22
Wheat straw	16.67	Ash	7.99
Premix	2.50	Calcium Ca	0.50
Total	100	Phosphorus P	0.35

## Data Availability

The data presented in this study are available in this article.

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
