# Peer review of "Effects of Duodenal 5-Hydroxytryptophan Perfusion on Melatonin Synthesis in GI Tract of Sheep"

_molecules, 2021, doi:10.3390/molecules26175275_

Round 1

Reviewer 1 Report

The authors properly agreed with the reviewer's observations, as well as correctly changed all the mistakes improving the article quality. Therefore, now I recommend this article for publication in Molecules journal.

Author Response

Thank you very much for your affirmation and recommendation of our manuscript to Molecules for publication.

Reviewer 2 Report

Authors should be commended for addressing previous reviews. 

Author Response

Thank you very much for your commendation, which will encourage us to work hard.

Reviewer 3 Report

The manuscript entitled “Effects of duodenal 5-hydroxytryptophan perfusion on melatonin synthesis in GI tract of sheep” had the objective of investigating the potential effects of 5-hydroxytryptophan (5-HTP) duodenal perfusion on melatonin synthesis in the gastrointestinal (GI) tract of sheep. The paper is very interesting and provides information about the peripheral synthesis of melatonin after intestinal infusion of 5-HTP and its contribution to melatonin concentration in different circulatory regions.

The manuscript is well written, but it has few sentences that should be rewritten for clarity. The conclusions should be provided to help to understand the paper. According to the journal instructions, the conclusion is not mandatory but can be added. Thus, the abstract should bring a conclusion besides results.

The M&M is not clear about the schedule of surgeries and sampling. How was it possible to perform many surgeries (catheter implantation and mucosal samples) and keep the schedule?

The units should be standardized. These forms are used in the text: nmol/g, ng/mL, pg/mg

L24- I suggest removing “(P<0.01)”. the P-value wasn’t presented after other results.

L73 – is there any reference for this affirmation?

L76-77- references for this affirmation should be provided

L79-81 – again same information without reference.

L128 “obviously” is not adequate or necessary in the sentence

L130 “pattern” seems to have a better fit than “tendency” in the sentence

L181 – the sentence “these levels was stable at the range” is not clear. Try rewriting it.

L212 – what can explain the difference in 50 times between studies?

L283- which factor was used for blocking?

L284 – treatment description should be clearer. What is treatment 2? Experimental group I and II is not an adequate name for treatment. It would be better a name directly related to the treatments. How were the 5-HTP doses chosen?

L324 -Which day the mucosal sample was collected? Was the mucosa sampling at the same time as blood sampling?

L 321 and L341 – convert “r/min” to “x g” force. This is an international unit.

L343- how the protein was quantified? It is not clear if the mucosal content of 5-HTP or MT were expressed in grams of protein or tissue.

L348 – describe the repeated measure.

Figure 3. It isn’t clear from each circulation the data is from. Check the formatting of Fig3B

Author Response

    Thank you again to allow us to further respond to the your comments. We have carefully read the comments from you and all the comments are valuable to improve the quality of our manuscript. Following is our response to your comments point by point. We hope that the revised version is satisfied publishing standard.

Comment:

The manuscript entitled “Effects of duodenal 5-hydroxytryptophan perfusion on melatonin synthesis in GI tract of sheep” had the objective of investigating the potential effects of 5-hydroxytryptophan (5-HTP) duodenal perfusion on melatonin synthesis in the gastrointestinal (GI) tract of sheep. The paper is very interesting and provides information about the peripheral synthesis of melatonin after intestinal infusion of 5-HTP and its contribution to melatonin concentration in different circulatory regions.

Our response 

Thank you very much for your affirmation of our manuscript.

Comment:

The manuscript is well written, but it has few sentences that should be rewritten for clarity. The conclusions should be provided to help to understand the paper. According to the journal instructions, the conclusion is not mandatory but can be added. Thus, the abstract should bring a conclusion besides results.

Our response: 

Thanks for your comments. We have added the simple conclusions both in the sections of abstract and discussion.

Comment:

The M&M is not clear about the schedule of surgeries and sampling. How was it possible to perform many surgeries (catheter implantation and mucosal samples) and keep the schedule?

Our response:

Thanks for your comment. We have added a diagram to describe the schedule you mentioned. (shown in figure 10)

Figure 10. Blood and mucosa sample collection procedure 

Comment:

The units should be standardized. These forms are used in the text: nmol/g, ng/mL, pg/mg

Our response

    We have carefully checked the units of tables and figures, one mistake was made in figure 2, where 5-HTP unit presented nmol/g was incorrect, correct unit was ng/mg. We tried to standardize all the units, such as ng/mg for mucosal 5-HTP and MT content, and ng/mL for blood 5-HTP and MT concentration.

Comment:

L24- I suggest removing “(P<0.01)”. the P-value wasn’t presented after other results.

Our response

    We have removed the “(P<0.01)” at the end of abstract.

Comment:

L73 – is there any reference for this affirmation?

Our response:

   Yes, we have added references [9,38].

Comment:

L76-77- references for this affirmation should be provided

Our response:

   Yes, reference [10] have been added.

Comment:

L79-81 – again same information without reference.

Our response:

   The references [9,38] have been added.

Comment:

L128 “obviously” is not adequate or necessary in the sentence

Our response

   The word “obviously” has been deleted.

Comment:

L130 “pattern” seems to have a better fit than “tendency” in the sentence

Our response

   The word “tendency” has been replaced by “pattern”.

Comment:

L181 – the sentence “these levels was stable at the range” is not clear. Try rewriting it.

Our response

    The sentence has been rewritten, as follow: these levels fluctuated in the range of 48.93 to 63.53 pg/mL during the entire perfusion period until to the 2 h after perfusion terminated.

Comment:

L212 – what can explain the difference in 50 times between studies?

Our response

    This difference may be due to 5-HTP orally administration, even if rumen-protected 5-HTP was used, whereas the effective dose of 5-HTP reaching the duodenum still unknown, because of the complicated rumen environment and the quality difference of rumen-protected products of some companies.

Comment:

L283- which factor was used for blocking?

Our response

    The sentence was changed as follow: a single factor randomized block design based on the sheep body weight.

Comment:

L284 – treatment description should be clearer. What is treatment 2? Experimental group I and II is not an adequate name for treatment. It would be better a name directly related to the treatments. How were the 5-HTP doses chosen?

Our response

    Yes, we have changed the name to 5-HTP treated group 1 and 2.

Comment:

L324 -Which day the mucosal sample was collected? Was the mucosa sampling at the same time as blood sampling?

Our response

The mucosa sample was collected at the same day, but different time, shown in figure 10.

Figure 10. Blood and mucosa sample collection procedure 

Comment:

L 321 and L341 – convert “r/min” to “x g” force. This is an international unit.

Our response

    We have changed all units “r/min” to “x g” force

Comment:

L343- how the protein was quantified? It is not clear if the mucosal content of 5-HTP or MT were expressed in grams of protein or tissue.

Our response

    We have addressed the sentence as follow: One gram protein of each sample was used for plasma and intestinal mucosa MT and 5-HTP determination, which were measured by enzyme immunoassay using an ELISA kit (Nanjing Jiancheng Bioengineering Institute, Nanjing, China) analyzed with ELISA analyzer (Waldron dr-200 Bs) following the manufacture’s instructions. This has been added into the text

Comment:

L348 – describe the repeated measure.

Our response

    We have addressed “repeated measure” as “repeated measure ANOVA”

Comment:

Figure 3. It isn’t clear from each circulation the data is from. Check the formatting of Fig3B

Our response

We have redrawn the figure 3 to make it clear.

This manuscript is a resubmission of an earlier submission. The following is a list of the peer review reports and author responses from that submission.

Round 1

Reviewer 1 Report

The present paper evaluates the effects of duodenal 5-hydroxytryptophan perfusion on melatonin synthesis in gastrointestinal tract of sheep.

Although the use of this approach could be an interesting way to increase the melatonin production in animals, the rationale of the study is not well supported and the paper is quite poorly presented with some inaccuracies, oversimplifications and mistakes. Therefore, I think that the paper does not provide enough relevant data to be considered for publication in Molecules.

Reviewer 2 Report

The current manuscript by Pan et al. studies the synthesis of melatonin in the GI tract following the administration of 5-HTP in sheep. The study is well conducted and provides evidence that the GI tract is a significant contributor to the overall production and release of melatonin that reaches the systemic circulation. They also provide evidence that the brain releases melatonin within the bloodstream due to concentration differentials of melatonin between the carotid artery and jugular vein. The manuscript provides new information to the field and the methods are adequate. There are a few points of clarification/edits that should be made prior to publication however.

  1. The authors state that one of their aims was to answer: “whether the GI synthesized melatonin will impact normal melatonin circadian rhythm.”

            The studies as shown in the manuscript are short in nature and do not reflect the totality of melatonin rhythms. The studies include data for up to 8 hrs after 5-HTP administrations, however this does not truly give the full picture as to what effect that would have over an entire circadian period (i.e. at least 1 day). This statement should be removed from the introduction.

  1. In lines 169 – 175, there are a few spots where 5-HTP is written as “5-HIP.” Please correct.
  2. Statistical analysis of the timecourses for melatonin concentrations should be a repeated measures analysis. This will strengthen the author’s analysis of their data and significance of their findings based on the data presented.

Reviewer 3 Report

The manuscript identified as “molecules-1254096” deals with the effects of duodenal 5-hydroxytryptophan perfusion on melatonin synthesis in GI tract of sheep. The topic is interesting and relevant, however major changes are needed to make this manuscript acceptable for publication in Molecules journal.

General Comments:

According to the authors, the aim of this study was to verify whether duodenal 5-HTP supplementation 78 will increase the GI melatonin synthesis; contribution of the GI synthesized melatonin to the circulatory melatonin; and verify whether the GI synthesized melatonin will impact the normal melatonin circadian rhythm. The Molecules journal covers high quality experimental and theoretical results, as well as well conducted studies. Therefore, there are some important issues, which need to be addressed:

1) It is a well-developed study, well-written version, and with relative scientific quality. However, the authors should improve the study novelty and benefits in the text (make them clear), since there are others studies covering the respective subjective, as follow:

- Rumen-Protected 5-Hydroxytryptophan Improves Sheep Melatonin Synthesis in the Pineal Gland and Intestinal Tract, DOI: 10.12659/MSM.915909;

- Effects of rumen protected 5-hydroxytryptophan on contents of 5-hydroxytryptophan, melatonin in gastrointestinal tract digesta and plasma of sheep, Chinese Journal of Animal Nutrition 2018 Vol.30 No.10 pp.4037-4047.

2) The introduction needs some work to better set up the stage for the aims of the study. It will help the author to better explain their results.

3) The authors properly presented the statistical results from the samples variability. However, the authors should improve the interpretation of the results clearly describing the relationship among them. A multivariate statistical analysis clearly describes the relationship among the experimental variables and samples characteristics.

If the authors do not develop the multivariate statistical analysis, they should improve the discussion covering the relationship among samples and variability.